# Outcomes of the Tall-Cell Variant of Papillary Thyroid Carcinoma in Patients with Different Ages: A 17-Year Mono-Institutional Experience

**DOI:** 10.3390/cancers15072152

**Published:** 2023-04-05

**Authors:** Agnese Proietti, Francesca Signorini, Riccardo Giannini, Anello Marcello Poma, Elisabetta Macerola, Liborio Torregrossa, Gabriele Materazzi, Alessio Basolo, Ferruccio Santini, Rossella Elisei, David Viola, Fulvio Basolo, Clara Ugolini

**Affiliations:** 1Section of Pathology, University Hospital of Pisa, 56126 Pisa, Italy; 2Department of Surgical, Medical, Molecular Pathology and Critical Area, University of Pisa, 56126 Pisa, Italy; 3Unit of Endocrinology, Department of Clinical and Experimental Medicine, University of Pisa, 56126 Pisa, Italyferruccio.santini@unipi.it (F.S.);; 4Section of Endocrinology, Versilia Hospital, 55041 Camaiore, Italy

**Keywords:** thyroid carcinoma, papillary carcinoma, tall cell, age, clinical outcome

## Abstract

**Simple Summary:**

This is the largest mono-institutional study addressing the debated issue about the impact of tall-cell areas on papillary thyroid carcinoma (PTC) prognosis correlated with patient age. The results of this series confirm that the tall-cell morphology alone in PTCs does not have the same negative prognostic significance in the younger population as in the older population.

**Abstract:**

The tall-cell variant of papillary thyroid carcinoma (TCPTC) is the most common aggressive variant of papillary thyroid carcinoma (PTC) and typically occurs in older patients. In this study, we analyzed retrospectively the largest mono-institutional series of PTCs with tall-cell features (989 patients) over a 17-year period, re-evaluating tumors based on age at presentation and outcomes in different age groups. We divided patients into three age groups following different criteria (the criterion from the American Joint Committee on Cancer Tumor Node Metastasis (AJCC TNM) guidelines, criterion for the statistical division into tertiles and adolescent/post-adolescent criterion) to analyze the clinicopathological characteristics in different age groups, especially in terms of recurrence-free survival (RFS) and distant recurrence-free survival (DRFS). We obtained three main results: 1. the population is distributed among the different age groups, and therefore, this type of cancer is not exclusively found among those of an older age; 2. in the RFS analysis, we can see a higher probability of local recurrence in the younger and older groups and, unexpectedly, a lower probability of local recurrence in the “median age” group; and 3. in the DRFS analysis, we can observe a higher probability of distant recurrence in older patients. From a molecular perspective, no significant differences in the mutational status of *BRAF* were detected according to different age groups, while mutations in the *TERT* promoter were exclusively present in older patients of all age groups, highlighting the potential prognostic implications of *TERT* promoter mutations in PTCs. In conclusion, the results of this series confirm that TC morphology alone in PTCs does not have the same negative prognostic significance in the younger population as in the older population. The reason for these different outcomes remains unclear and needs further studies.

## 1. Introduction

Papillary thyroid carcinoma (PTC) is the most common endocrine neoplasia. The incidence of PTC is on the rise worldwide, and this disease is projected to become the fourth leading type of cancer across the globe. From 1990 to 2013, the global age-standardized incidence rate of thyroid cancer increased by 20%. This global rise in incidence has been attributed to several factors, including the increased detection of early tumors, the elevated prevalence of modifiable individual risk factors (for example, obesity) and the increased exposure to environmental risk factors (for example, iodine levels) [1]. Papillary thyroid carcinomas are usually associated with good survival; however, some variants of papillary thyroid carcinomas may behave more aggressively than classical papillary thyroid carcinomas (CPTCs). Several histological variants associated with a different rate of aggressive features and clinical behavior are well recognized. Among the aggressive variants, the tall-cell variant of PTC (TCPTC) is the most common, with a prevalence of up to 12% of all PTCs [2,3]. It is morphologically characterized by tall cells with a height that is at least three times the width, and it is reported that this variant typically occurs in older patients [4]. Age at diagnosis has been identified as one of the most important prognostic factors in PTCs. The prognosis of patients younger than 55 years old has been demonstrated to be better than that of patients 55 years old or older, with a 10-year disease-specific survival (DSS) rate that exceeds 99% [5,6,7]. For this reason, the 8th edition of the American Joint Committee on Cancer (AJCC) staging system used a cut-off age of 55 years to stratify risk [8], which was an increase in the cut-off value of 10 years compared to that used in the last (7th) edition of the AJCC staging system [9]. Recently, many studies analyzing the morphological features and the percentage of tall cells and their correlations with outcomes have been published [10,11,12,13], with important consequences for diagnosis and classification [4]. Although the survival of patients <55 years old with PTC is excellent, it is still unclear whether patients <55 years old with TCPTC have the same excellent outcome compared to the older group (≥55 years). Up to now, very few studies have analyzed the prognosis of TCPTCs in different age groups.

To understand the differences in the outcomes of TCPTC patients with different ages, we collected a large mono-institutional series of TCPTC data by retrospectively selecting cases of any age and with any tall-cell area from over a 17-year period.

## 2. Materials and Methods

### 2.1. Sample Selection

All PTCs with any TC component diagnosed between 2001 and 2017 at the Unit of Surgical Pathology of the University Hospital of Pisa were included in this study. All patients underwent total thyroidectomy at the Unit of Surgical Pathology of the University Hospital of Pisa and, whenever appropriate, lymph node dissection and radioiodine treatment according to the American Thyroid Association (ATA) guidelines [14]. For each case, glass slides were retrieved and re-evaluated independently by two pathologists (C.U. and F.B.) using the 2017 WHO (World Health Organization) criteria [4]. Tumors with aspects of other aggressive variants, such as hobnail, columnar and solid/trabecular aspects, were excluded. All cases were re-staged according to the latest edition of the American Joint Committee on Cancer (AJCC) staging system. Age at diagnosis and other clinicopathological features were assessed for each patient, including gender, tumor size, neoplastic capsular infiltration, thyroid parenchyma infiltration, extra-thyroid extension (ETE), multifocality, bilaterality, lymph node metastasis, vascular embolization and follow-up data, particularly regarding recurrence-free survival (RFS) and distant recurrence-free survival (DRFS). Recurrence-free intervals were defined as the time in years from surgical resection to the date of local or distant recurrences. Cases without recurrence were right-censored. This study was approved by the ethics committee, “Comitato Etico di Area Vasta Nord Ovest” (CEAVNO) for Clinical Experimentation (protocol number: ID19211), and informed consent was given by each patient.

### 2.2. Age Grouping of TCPTC Patients

All patients were divided based on age at diagnosis, following three different criteria: 

First criterion, following the AJCC guidelines (AgeAJCC): patients < 55 years old and ≥55 years old;

Second criterion, following the statistical division into tertiles (AgeNORM): patients ≤3 8 years old, 39–57 years old and ≥58 years old;

Third criterion, the following adolescent/post-adolescent criterion [15,16] (AgeCOD): patients ≤ 25 years old and >25 years old.

### 2.3. Genotyping

For a subgroup of patients with follow-up data, the mutational status of *BRAF* (exon 15) and the *TERT* promoter was investigated. For each case, manual macro-dissection was performed. DNA was purified from two unstained, formal-fixed, paraffin-embedded, 10 µM thick sections using the QIAmp DNA mini kit (Qiagen, Hilden, Germany). Direct sequencing was performed according to standard procedures on a 3130 Genetic Analyzer (Thermo Fisher, Waltham, MA, USA).

### 2.4. Statistical Analyses

The Shapiro–Wilk test was performed to assess the normality of the distribution of the age within the study group. Survival curves were built with the Kaplan–Meier method and compared via the log-rank test. The Cox proportional hazards regression was used to calculate hazard ratios and their associated 95% confidence interval (95% CI) to estimate the risk of local or distant recurrences for the age groups and for gender, tumor size, neoplastic capsular infiltration, thyroid parenchyma infiltration, extra-thyroidal extension, multifocality, bilaterality, lymph node metastasis and vascular embolization.

## 3. Results

### 3.1. Study Population

A total of 3092 PTC patients with a tall-cell area were examined, of which 2300 were diagnosed with TCPTC according to the 2017 WHO criteria. Among those with follow-up data, 989 patients were eligible for inclusion in the study, with a median follow-up of 5.6 years. The mean age was 47.7 years old (range of 12–86). The mean size of the neoplasms was 1.52 cm (range of 0.2–9 cm). In total, 774 patients (78.3%) were female. Overall, 91 patients (9.2%) experienced local recurrences while 24 patients (2.4%) developed distant recurrences or metastases.

### 3.2. Clinicopathological Features in Different Age Groups

As previously mentioned, patients were divided into three age groups.

In the first group (<55 years old and ≥55 years old, according to the AJCC criterion), 691 patients were younger than 55 years old and 298 were older. There were 54 (7.8%) local recurrences in patients <55 years old and 37 (12.4%) in patients ≥55 years old. There were 6 (0.9%) distant recurrences in the <55 group and 18 (6.0%) in the ≥55 group (Table 1 and Table 2). Analyzing the HRs for the association between age classes and DRFS, we found a significant difference between younger and older patients with a higher probability of events in the older class (*p* < 0.001) (Figure 1, Table 2). No significant differences in the mutational status of *BRAF* were detected in the two groups; however, mutations in the *TERT* promoter were exclusively present in older patients (Table 1 and Table 2).

In the second group, following the statistical division into tertiles, the population was divided into a lower group that was ≤38 years old (N = 242), a median group that was 39–57 years old (N = 516) and an upper group that was ≥58 years old (N = 231). Local recurrences were: 35 (14.5%), 21 (4.1%) and 35 (15.2%). Distant recurrences were 0 (0%), 6 (1.2%) and 18 (7.8%) (Table 1 and Table 2). Analyzing the HRs for the association between age classes and RFS, we found a significant difference between lower, median and older patients with a higher probability of events in the lower and upper age classes, while the median age class showed a lower probability (*p* < 0.001) (Figure 1, Table 2). Analyzing the HRs for the association between age classes and DRFS, we identified a significant difference between median and older patients with a higher probability of events in older patients (*p* = 0.000000006) (Figure 1, Table 1). Since no distant recurrence events occurred in the youngest group, the hazard ratio could not be estimated (Table 1 and Table 2). No significant differences in the mutational status of *BRAF* were detected in the three groups; however, mutations in the *TERT* promoter were exclusively present in older patients (Table 1 and Table 2).

In the third group (≤25 years old and >25 years old, following the adolescent/post-adolescent criterion), 59 patients were less or equal to 25 years old (6%) and 930 were older (94%). There were 17 (28.8%) local recurrences in the younger group, and 74 (8.0%) in the older one. There were 24 (2.6%) distant recurrences, all of which were in the older group (Table 1 and Table 2). Analyzing the HRs for the association between age classes and RFS, we found a significant difference between younger and older patients, with a higher probability of events in the younger age class (*p* < 0.001) (Figure 1, Table 2). Since no distant recurrence events occurred in the younger group, the hazard ratio could not be estimated (Table 1 and Table 2). No significant differences in the mutational status of *BRAF* was detected in the two groups; meanwhile, mutations in the *TERT* promoter were exclusively present in the older group (Table 1 and Table 2).

## 4. Discussion

The tall-cell variant is recognized as an aggressive variant of PTC [4,17,18].

Only a few patients with PTC are affected by a clinically aggressive tumor. One of the most common subtypes associated with disease persistence and recurrence is TCPTC. Currently, this variant is underdiagnosed: several studies have demonstrated that when cases diagnosed as CPTC were reviewed by endocrine pathologists, 1–13% of the tumors were identified as TCPTCs. One of the obstacles for the correct diagnosis of TCPTC is the lack of consensus as to how much of the tumor must be composed of tall cells to make a diagnosis [18].

By definition, the TCPTC variant is composed of tall cells with a height that is at least three times the width, with abundant eosinophilic (oncocytic-like) cytoplasm, typical nuclear features of PTC and nuclear pseudoinclusions. Because tall-cell areas are frequently present in otherwise conventional papillary carcinomas, tall cells must account for ≥30% of all tumor cells to diagnose the tall-cell variant [4].

There are serious prognostic and management implications around a diagnosis of TCPTC, and most authors believe that one of the reasons for the worse prognosis of TCPTC is related to its advanced age at presentation [19].

In a recent study, Sugino et al. evaluated the outcomes of differentiated thyroid cancers in relation to age groups. Although the number of patients <15 years old was small, cN1, ETE, a large primary tumor size and synchronous distant metastases were observed more frequently in the younger age group. This indicates that younger patients would have poor disease-free survival (DFS) and distant metastasis-free survival (DMFS) as determined by univariate analysis; however, an age cut-off has not been identified from a treatment outcome perspective [20].

In previous studies, the age at presentation of pediatric differentiated thyroid cancers with lymph node metastases or synchronous distant metastases was younger than that of children without metastases at the time of presentation, despite having a similar tumor size, and, similarly, children who developed metastases during follow-up were younger than those who did not [21]. 

The mean age of TCPTC at presentation is reported to be older than that of classical variants of PTCs (CPTCs), with a range of TCPTC patients between 41 and 66 years old versus a range of CPTC patients between 34 and 53 years old [22]. Moreover, different database studies have shown that the age at presentation of TCPTC patients was generally older than that of CPTC patients (55.3 years old vs. 47.1 years old) [17,23]. Although TCPTC is exceptional in children, cases of TCPTCs are reported in the pediatric and adolescent population [24].

In PTCs, genetic alterations have been found in about 95% of tumors: *BRAF* mutations and *RET* rearrangements are the main genetic drivers. The occurrence of secondary mutations is typical of poorly differentiated thyroid carcinoma and anaplastic thyroid carcinoma. Although less common, secondary mutations have been found also in well-differentiated tumors. Among them, telomerase reverse transcriptase (*TERT*) promoter mutations are present in more advanced tumors as well as in differentiated ones. These mutations are of particular interest since they are associated with a poor outcome, especially when co-occurring with *RAS* or *BRAF* mutations. Nevertheless, their usefulness as an independent poor prognostic marker is still questioned due to their occurrence in tumors with aggressive pathological features and in older patients [25].

Recent studies have analyzed the molecular characteristics related to TCPTCs, highlighting how the coexistence of *BRAF* and *TERT* promoter mutations may be associated with aggressive pathological features, as compared with tumors with a single mutation or no mutations in these genes. In particular, *BRAF* and *TERT* promoter co-mutations were more commonly detected in male patients at an older age, that had a larger tumor size and were more prone to extra-thyroid invasion [26]. A *TERT* promoter mutation may have important prognostic implications as a strong predictor of tumor relapse, being associated with the development of distant metastases in PTCs [27,28].

In this study, we analyzed retrospectively the largest mono-institutional series of PTCs with tall-cell features over a 17-year period, re-evaluating tumors based on age at presentation and outcomes in different age groups. We analyzed a total of 989 patients with TCPTCs and follow-up data (median follow-up of 5.6 years) from a single institution. We divided the population into three age groups following different criteria (AgeAJCC criterion: patients < 55 years old and ≥55 years old; AgeNORM criterion, following statistical division into tertiles: patients ≤ 38 years old, 39–57 years old and ≥58 years old; and AgeCOD criterion, following adolescent/post-adolescent classification: patients ≤ 25 years old and >25 years old). 

The first finding is that the TCPTC population is distributed among the different ages, with a percentage of adolescent patients (≤25 years old, N = 59) of 6% and a percentage of pediatric patients (<18 years old, N = 13) of 1.3%, which is virtually identical to the number of patients over 70 years old (N = 58): therefore, this type of cancer is not exclusively found in patients of an older age.

Follow-up data demonstrate a dual aspect of disease outcomes. First, in the RFS analysis we can see a higher probability of local recurrence in the younger and older groups and, unexpectedly, a lower probability of local recurrence in the “median age” group. This is particularly evident following the criterion of the age grouping into tertiles, in which three categories were identified. In the other two age-grouping categories, a lower probability of local recurrence is observed in the older groups, probably because the older groups include the “median age” patients.

Secondly, in the DRFS analysis, the scenario changes significantly. Indeed, we can observe a higher probability of distant recurrence in older patients in all age-grouping categories. Evaluating other clinicopathological data, we can assert that aggressive disease is driven by classical clinicopathological features of aggressiveness: in multivariate analysis, extrathyroidal infiltration and *TERT* promoter mutational status show the strongest association for RFS and DRFS, while a larger tumor size plays an important role even if only in association with RFS. Our study has already analyzed a large group of TCPTC patients, obtaining equivalent results regardless of the different percentages of tall cells.

From a molecular perspective, no significant differences in the mutational status of *BRAF* were detected according to the different age groups, while mutations in the *TERT* promoter were exclusively present in older patients of all age groups, highlighting the potential prognostic implications of *TERT* promoter mutations in PTCs.

The most significant limitation of this study lies in the small number of younger patients (≤25 years old) with TCPTCs; however, the number of adolescent patients (N = 59) included in the analysis is higher than that in similar studies published before.

## 5. Conclusions

In conclusion, this retrospective study of TCPTCs, based on age-grouping categories (AgeAJCC criterion, AgeNORM criterion and AgeCOD criterion) showed results that can be summarized in three main points.

First, our analysis showed that the population is distributed among the different age groups and TCPTC is present even in young and very young patients; therefore, this type of cancer is not exclusively found in patients of an older age. Secondly, in the RFS analysis, we can see a higher probability of local recurrence in the younger and older groups and, unexpectedly, a lower probability of local recurrence in the “median age” group. Finally, in the DRFS analysis, we can observe a worse prognosis with regard to distant recurrence in older patients.

No significant differences in the mutational status of *BRAF* were detected according to the different age groups, while mutations in the *TERT* promoter were exclusively present in older patients of all age groups, highlighting the potential prognostic implications of *TERT* promoter mutations in PTCs.

The results of this series confirm that TC morphology alone in PTCs does not have the same negative prognostic significance in the younger population as in the older population. The reason for these different outcomes is, up to now, unclear and needs further large, database-initiated, multicentric studies in order to obtain a more clear view of this tumor.

## Figures and Tables

**Figure 1 cancers-15-02152-f001:**
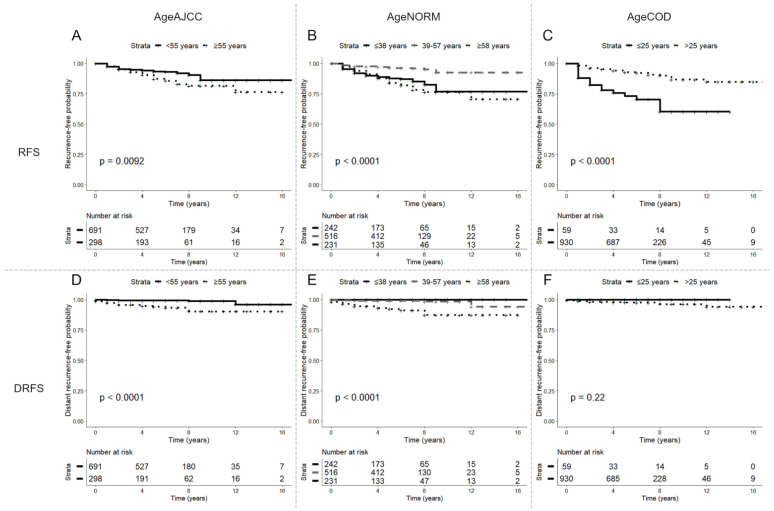
Kaplan–Meier curves according to age groups (AgeAJCC criterion: patients < 55 years old and ≥55 years old; AgeNORM criterion, following statistical division into tertiles: patients ≤ 38 years old, 39–57 years old and ≥58 years old; AgeCOD criterion, following adolescent/post-adolescent classification: patients ≤ 25 years old and >25 years old). (**A**–**C**) Recurrence-free survival; (**D**–**F**) distant recurrence-free survival. (**A**–**D**) Stratification by AJCC age cut-off; (**B**–**E**) stratification by tertiles; (**C**–**F**) stratification by adolescent/post-adolescent classification.

**Table 1 cancers-15-02152-t001:** Clinicopathological data for all age-grouping categories (AgeAJCC criterion: patients < 55 years old and ≥55 years old; AgeNORM criterion, following statistical division into tertiles: patients ≤ 38 years old, 39–57 years old and ≥58 years old; AgeCOD criterion, following adolescent/post-adolescent classification: patients ≤ 25 years old and >25 years old) (* *BRAF* and *TERT* promoter mutational status was assessed in a subgroup of patients).

		AgeNORM				AgeAJCC			AgeCOD		
		Lower	Median	Upper	*p*	<55	≥55	*p*	≤25	>25	*p*
Total (989)		242	516	231		691	298		59	930	
Gender	F	196	409	169	0.092	553	221	0.043	46	728	1
	M	46	107	62		138	77		13	202	
Extra-thyroid extension	NO	113	211	73	0.003	294	103	0.019	29	368	0.17
	YES	129	305	158		397	195		30	562	
Multifocality	NO	138	253	105	0.032	355	141	0.26	34	462	0.28
	YES	104	263	126		336	157		25	468	
Bilaterality	NO	174	320	136	0.006	450	180	0.17	40	590	0.57
	YES	68	196	95		241	118		19	340	
Node metastasis	NO	163	397	182	0.0065	505	237	0.037	35	707	0.0076
	YES	79	119	49		186	61		24	223	
Embolization	NO	184	402	176	0.79	533	229	0.93	41	721	0.15
	YES	58	114	55		158	69		18	209	
TNM size	T1a	65	213	68	0.0000003	247	99	0.0033	8	338	0.0011
	T1b	128	228	97		329	124		36	417	
	T2	47	71	56		109	65		15	159	
	T3a	2	4	10		6	10		0	16	
BRAF mutation *	NO	21	15	24	0.41	35	25	0.17	13	47	0.17
	YES	82	59	138		133	146		38	241	
TERT mutation *	NO	62	0	103	0.002	62	103	0.008	35	130	0.08
	YES	1	0	19		1	19		1	19	
RFS	NO	207	495	196	0.00000004	637	261	0.029	42	856	0.000006
	YES	35	21	35		54	37		17	74	
DRFS	NO	242	510	213	0.000000006	685	280	0.000006	59	906	0.39
	YES	0	6	18		6	18		0	24	

**Table 2 cancers-15-02152-t002:** Clinicopathological data distribution for all age-grouping categories with hazard ratio (HR) analysis (AgeAJCC criterion: patients < 55 years old and ≥55 years old; AgeNORM criterion, following statistical division into tertiles: patients ≤ 38 years old, 39–57 years old and ≥58 years old; AgeCOD criterion, following adolescent/post-adolescent classification: patients ≤ 25 years old and >25 years old) (* baseline for ° hazard ratio (HR) calculation; *** BRAF* and *TERT* promoter mutational status was assessed in a subgroup of patients; NA = not available).

			RFS				DRFS			
		Total	nr	%	HR °	*p*	nr	%	HR °	*p*
Gender	Male *	215	57	26.5	2.2 (1.4–3.4)	<0.001	10	4.7	2.7 (1.2–6)	0.019
	Female	774	34	4.4			14	1.8		
Extra-thyroid extension	No *	397	17	4.3	3 (1.8–5.1)	<0.001	2	0.5	7.6 (1.8–32)	0.006
	Yes	592	74	12.5			22	3.7		
Multifocality	No *	496	33	6.7	1.9 (1.2–2.9)	0.004	7	1.4	2.6 (1.1–6.3)	0.033
	Yes	493	58	11.8			17	3.4		
Bilaterality	No *	630	47	7.5	1.7 (1.1–2.6)	0.012	9	1.4	3 (1.3–6.9)	0.008
	Yes	359	44	12.3			15	4.2		
Node metastasis	No *	742	37	5.0	5.1 (3.4–7.8)	<0.001	8	1.1	6.7 (2.9–16)	<0.001
	Yes	247	54	21.9			16	6.5		
Embolization	No *	762	64	8.4	2 (1.3–2.2)	0.003	18	2.4	1.5 (0.5–3.7)	0.435
	Yes	227	27	11.9			6	2.6		
TNM size	T1a *	346	12	3.5			11	3.2		
	T1b	453	30	6.6	2 (1–3.8)	0.05	5	1.1	0.3 (0.1–3.2)	0.05
	T2	174	39	22.4	6.8 (3.6–13)	<0.001	7	4.0	1.2 (0.4–3.2)	0.646
	T3a	17	6	35.3	36 (15.4–83.8)	<0.001	1	5.9	2.4 (0.3–19.3)	0.386
AgeNORM	Lower *	242	35	14.5			NA	NA	NA	NA
	Median *	516	21	4.1	0.2 (0.1–0.4)	<0.001	6	1.2	6.7 (2.6–16.6)	<0.0001
	Upper	231	35	15.2	1.1 (0.7–1.8)	0.507	18	7.8		
AgeAJCC	<55 *	691	54	7.8	1.7 (1.1–2.6)	0.01	6	0.9	7.5 (3–19)	<0.001
	≥55	298	37	12.4			18	6.0		
AgeCOD	≤25 *	59	17	28.8	0.2 (0.1–0.4)	<0.001	0	0.0	NA	0.392
	>25	930	74	8.0			24	2.6		
BRAF mutation **	No *	60	6	10.0	1.8 (0.76–4.2)	0.183	1	1.7	3.7 (0.5–28)	0.2
	Yes	279	44	15.8			16	5.7		
TERT promoter mutation **	No *	165	27	16.4	3.1 (1.4–6.8)	0.006	9	5.4	3.1 (0.8–11.6)	0.09
	Yes	20	8	40.0			3	15.0		

## Data Availability

The data analyzed in this study are available within the manuscript.

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
