# Peer review of "Outcomes of the Tall-Cell Variant of Papillary Thyroid Carcinoma in Patients with Different Ages: A 17-Year Mono-Institutional Experience"

_cancers, 2023, doi:10.3390/cancers15072152_

Round 1

Reviewer 1 Report

The authoring team for this manuscript has done a decent effort assessing the effects of age at diagnosis, on the severity and risk of progression of tall cell variant of papillary thyroid cancer, a tumor that is shown to be significantly more aggressive compared to the classical type of PTC. In addition they describe the effects of features of aggressiveness of the tumor with regard to recurrence free survival and distant recurrence free survival, and identify ETE and TERT promoter mutations as important predictive factors in that regard. A strength of this manuscript lies in the de-novo review of all the slides for these patients and their use of updated diagnoses according to the current standards, the prospective nature of follow up and the relatively large number of cases for this uncommon form of PTC.

The study design is appropriate, the statistical methods decent, and the results clear and concise.

Despite these positive geatures, this manuscript requires some minor changes, those including:

1. A more detailed review of the data on the effects of TERT promoter mutations on TCV PTC based on the currentl literature.

2. A mention of other mutations associated with higher risk TCV tumors based on the current literateure, their effects and their potential interactions with TERT promoter mutations.

3. A mention of the limitations of this study, including the fact that the younger patients with TCV are of a small number, even though higher than ever published before.

4. A mention of the need for large, database initiated, multicentric studies in order to obtain a more clear view of this tumor.

Author Response

Thank you for the comment and suggestions, which we have taken to make changes to the article.

In particular,

  1. in the discussion, TERT promoter mutations in PTCs and specifically in TCPTCs based on the current literature has been deepened;
  2. in the discussion, other mutations present in PTCs and TCPTCs were examined based on the current literature, in order to show their role in these tumors also in association with TERT promoter mutations;
  3. in the discussion, limitations of the study were added, in particular by referring to the number of young patients (≤25 years) involved in the study;
  4. in the conclusions, we hope for the need for further large multicentric studies to further deepen the prognostic significance of TC morphology in the younger as in the older population.

Reviewer 2 Report

This is a very interesting study on an important and very debated topic. The sample is large and the study very well conducted. The results are very important and can change the approach on these patients.

I think limitations of the study can be better discussed.

References can be improved (see for example Longheu A et al. Tall Cell Variant versus Conventional Papillary Thyroid Carcinoma: A Retrospective Analysis in 351 Consecutive Patients. J Clin Med. 2020 Dec 28;10(1):70.

Author Response

Thank you for the comment and suggestions, which we have taken to make changes to the article.

In particular,

  1. in the discussion, limitations of the study were added, in particular by referring to the number of young patients (≤25 years) involved in the study;
  2. references have been improved [24-28].

Reviewer 3 Report

A well-planned study. Only minor comments:

TERT promoter mutation was present exclusively in the older age group. As this may have significant prognostic implications, this should be added to the abstract and conclusions. Also whether this mutation can be responsible for the more frequent distant metastases in that age group needs to be discussed (even if not significant in current study). Earlier reports have shown TERT promoter mutation to be associated with distant metastases (PMID: 25583906)

It would be good if the conclusions can be formulated as a paragraph instead of putting them point-wise.

Minor grammatical errors need to be looked into. For example, "Our study has already analyzed a numerous group of PTCs with different percentages of tall cells finding equivalent results", needs to be re-worded.

Author Response

Thank you for the comment and suggestions, which we have taken to make changes to the article.

In particular,

  1. TERT promoter mutations have been added to the abstract and in the conclusion in order to underline their potential prognostic implications in PTCs and TCPTCs. Moreover, in the discussion, TERT promoter mutations in PTCs and specifically in TCPTCs based on the current literature has been deepened, considering also the association between TERT promoter mutations and distant metastases;
  2. conclusions have been reformulated as paragraph;
  3. the article has been checked to identify and correct minor grammatical errors.